# Design dimensions of electrocutaneous warning stimulation patterns in workplace safety devices

**Eva-Maria Dölker** [1]☯*, **Stephan Lau**[1,2]☯, **Maria Anne Bernhard**[1], **Jens Haueisen**[1]

**1** Institute of Biomedical Engineering and Informatics, Technische Universität Ilmenau, Ilmenau, Germany,
**2** Australian Institute for Machine Learning (AIML), The University of Adelaide, School of Computer Science, Adelaide, Australia

☯ These authors contributed equally to this work.
* eva-maria.doelker@tu-ilmenau.de

**Data Availability Statement:** All data are in the manuscript and/or supporting information files at https://doi.org/10.5281/zenodo.12800471.

## Abstract

Warning of workers in dangerous situations is crucial. With the aim of deriving practical parameters for an electrocutaneous warning stimulation, we explore the design dimensions of pulse intervals, amplitudes, and locations of electrocutaneous stimulation in a study on healthy volunteers. Using biphasic rectangular current pulses on the upper right arm of 81 healthy participants, they evaluated temporal perception with varying intervals, ranging from 200 ms down to 0.5 ms, categorizing it as 'Individual pulses', 'Pulsating', 'Vibrating', or 'Continuous'. Next, we tested nine amplitude levels. Participants rated the perceived amplitude on a scale from 1 to 9 after a training phase. Finally, we presented five consecutive stimulation pulses in a pseudo-random order at eight electrode pair positions, asking participants to report the stimulated electrode pair. Participants perceived electrocutaneous pulses as 'Individual pulses' for median intervals above 74 ms, as 'Pulsating' between 44 ms and 74 ms, as 'Vibrating' between 12 ms and 44 ms, and as 'Continuous' below 12 ms. Pulse intervals below about 1 ms were perceived as weak and at about 5 ms as inconvenient, rendering these intervals less suitable for the design of a warning pattern. The median reported amplitudes [25%-75%-percentile] for presented amplitudes 1 to 9 are: 1[1–1], 2[2–3], 3[2–4], 3[3–4], 4[3–5], 5[4–6], 6[4–7], 7[5–8] and 7.5 [6–8] indicating a linear relationship between presented and perceived amplitude. These results suggest that the stimulation amplitude may be incorporated into a structured stimulation pattern. The majority of the electrode pair locations were reported correctly (64.3%–86.6%) or within the two neighboring electrode pairs (98%–99.7%). We conclude that the determined pulse intervals combined with the differentiability of locations offer the basis for designing a warning signal. Our research lays the groundwork for developing suitable signals for wearable electric warning devices.

## Introduction

In the realm of workplace safety, effectively alerting employees to hazards is crucial. Current methods like sound [1,2] or visual [3] cues can be ineffective in noisy or low-visibility

**Funding:** We would like to thank the Deutsche Forschungsgemeinschaft for its support under project number DFG-Ha2899/23-2 awarded to JH. The funders had no role in study design, data collection and analysis, decision to publish, or preparation of the manuscript. We acknowledge support for the publication costs by the Open Access Publication Fund of the Technische Universität Ilmenau.

**Competing interests:** The authors have declared that no competing interests exist.

conditions. Thus, our long-term goal is to create a new warning system using electrical stimulation as an additional channel for information transfer to workers in dangerous situations. The electrocutaneous stimulation for an electrical warning system involves delivering controlled electrical impulses to the skin, creating tactile sensations typically without causing pain or discomfort.

For our electrical warning system, we envisioned several functional requirements. Firstly, as part of the personal safety equipment, the electrical warning signal must be perceivable, distinguishable from other stimuli, and unique. Secondly, a warning system should be capable of transferring a small number of signal types. Thirdly, the electrodes of an electrical warning system should be placed on a part of the body where the wearing comfort is high, the electrodes are easy to attach, the amount of sweat is relatively low, the restriction of motion is minimal, the potential for conflict with working and personal safety equipment is low, and the risk of muscle contraction and unexpected involuntary responses is relatively low.

In addition to these functional requirements, an electrical warning system could adhere to the ISO EN 60601–1 norm [4], which defines safety and ergonomic requirements for medical electrical devices and systems, as there is currently no specific norm for electrical warning systems. Lastly, usability is a major requirement for our electrical warning system. It should be well-tolerated by the user, wearable for up to 8 hours, lightweight, integrated into work clothes, and equipped with stable electronics and a reliable energy supply.

Warning signals could convey varying meanings and the level of urgency. For instance, railway workers need reliable alerts about approaching trains, including track information and distance [1]. Thus, it is important to derive practical parameters for designing such electrocutaneous warning signals. Here, we explore the dimensions of pulse intervals, amplitudes, and locations of electrocutaneous stimulation in a study on healthy volunteers.

Noninvasive electrical stimulation is broadly applied in areas such as muscle stimulation [5–8] and transcutaneous electrical nerve stimulation (TENS) [9–11], as well as in specialized uses like electrical acupoint stimulation to reduce postoperative nausea and vomiting [12]. Recent studies on electrical muscle stimulation have explored the impact of muscle length on torque generation [6], its effects on visual sensory reweighting dynamics [5], and muscle metabolism assessment [7]. TENS has been studied for pain management [9,10] and reducing postoperative wound infections [11]. Electrocutaneous stimulation with information transfer is commonly utilized in medical prostheses to offer sensory feedback [13–16]. However, the purpose and consequently the parameters are different for electrocutaneous warning signals. For example, sensory feedback in medical prostheses uses considerably lower amplitudes than electrocutaneous warning [17]. Consequently, for the developmental process of an electrical warning system, parameter studies are needed.

In a preliminary study with 4 participants (f = 2, m = 2) [18], various stimulus and electrode parameters were tested on the upper right arm, revealing qualitative perceptions and three distinct thresholds: (1) The perception threshold is defined by the detection of a just noticeable stimulus; (2) The attention threshold is marked by a stimulus that draws attention to itself; (3) The intolerance threshold is reached when a stimulus generates perceptions that are intolerable. These thresholds were examined in a larger study involving 81 volunteers (f = 29, m = 52) [17], along with an investigation into the impact of pulse width, electrode size, and position. For a pulse width of 150μs, median values of 3.5 mA, 6.9 mA, and 13.8 mA were observed for perception, attention, and intolerance thresholds, respectively. Thresholds decrease with increasing pulse width. Lateral electrode positions have higher intolerance thresholds than medial ones, while perception and attention thresholds show no significant difference across positions. Electrode size (15×15 mm$^2$ to 40×40 mm$^2$) does not notably affect the thresholds. Knocking is the predominant sensation at perception and attention thresholds, while muscle

twitching, pinching, and stinging are common at the intolerance threshold. Biphasic stimulation pulse widths between 150 μs and 250 μs are recommended for electric warning wearables [17].

Previous research also investigated the suitability of varying current stimulation amplitudes, pulse intervals, and stimulation locations within the field of neurofeedback. Parameters used for encoding information are e.g. pulse frequency [19,20], number of pulses [21], and pulse width [19] whereas a fixed amplitude was often used. The spatial discriminability depends on the temporal stimulation parameters [22] and spatial identification was improved if only one electrode pair was stimulated instead of two simultaneously [21].

Building on these findings, our current research explored the effectiveness of varying current stimulation amplitudes, pulse intervals, and stimulation locations specifically for providing warning information. These parameters need to be investigated separately since the application in electrical warning systems has different requirements compared to neurofeedback applications. This work contributes to the baseline studies as part of the developmental process of an electrical warning system.

## Methods

### Study group

Table 1 presents the descriptive statistics of the study group. The recruitment period of this study was from 27.11.2018 until 19.06.2019. Approval for the study was granted by the ethics committee of the Faculty of Medicine at Friedrich-Schiller-University Jena, Germany. All procedures were conducted in compliance with applicable guidelines and regulations, and all participants provided written informed consent.

On the day of the experiment and the preceding day, participants were instructed to ensure they obtained an ample amount of sleep, refrained from consuming caffeine, nicotine, or alcohol, drank a sufficient quantity of fluids (approximately 2 liters), avoided engaging in strenuous physical activities or sports, and refrained from applying skin cream to their upper arms. Further exclusion criteria were heart disease or pacemaker, severe high blood pressure or hyperthyroidism, neurological or psychiatric pre-existing conditions, diabetes, numbness in the upper arm, implants in the right arm, significant skin diseases on the right upper arm, current or previous abuse of addictive substances, pregnancy or breastfeeding, and regular use of medication except contraception.

### Experimental setup

The experimental setup (Fig 1) was identical to the one in our previous publication [17]. This paper contains a second set of experiments. Here, we give a brief overview of the experimental setup, for details please see [17].

**Table 1. Descriptive statistics of the study group.**

| Property | Quantity |
|---|---|
| Participants | $n = 81$ |
| Gender | female: 29, male: 52 |
| Age in years | 27±7.8 <br> youngest: 20, oldest: 52 |
| Handedness | right-handed: 72, left-handed: 9 |
| Arm circumference in cm | 30.3±4.4 |

Age and arm circumference are given as mean values±standard deviation.

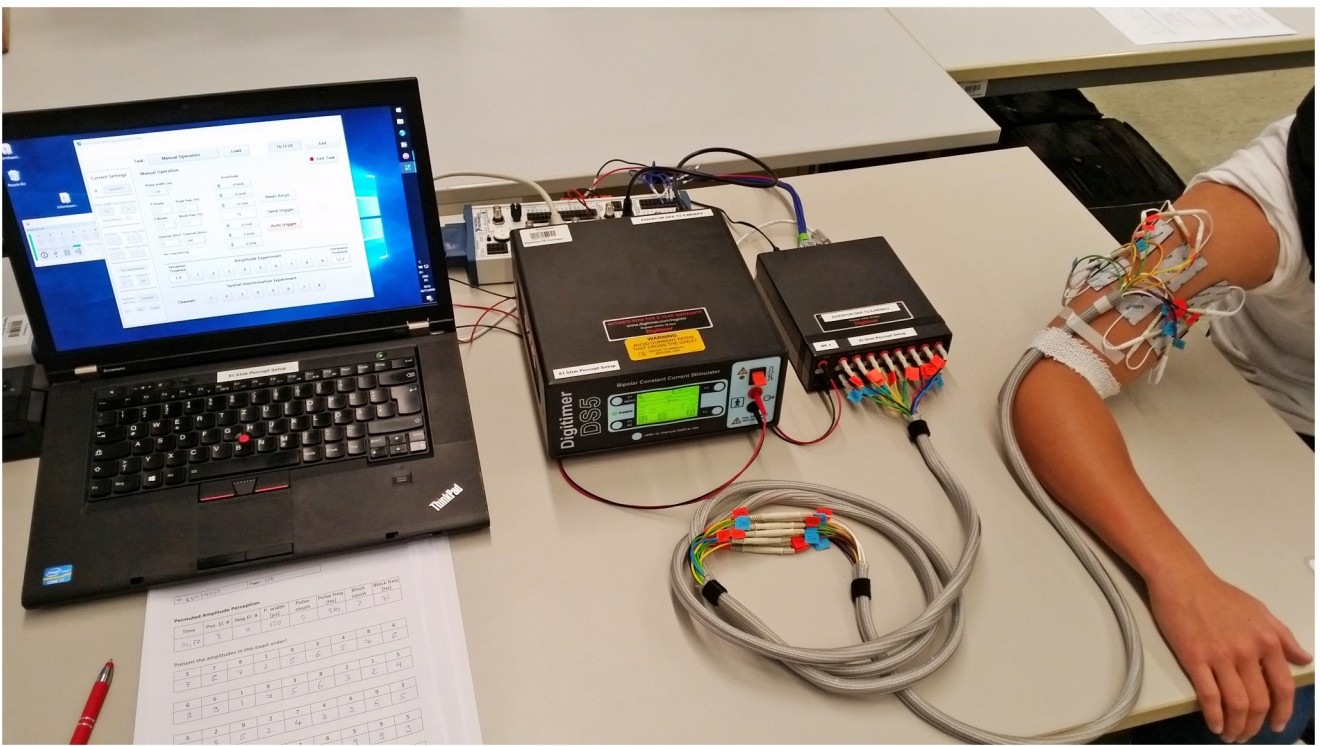

**Fig 1. Photograph of the experimental setup.** Left to right: Computer running LabVIEW, constant current stimulator DS5, multiplexer D188, and electrodes at the upper arm of one volunteer.

We employed a custom-built program developed in LabVIEW 2017 (National Instruments, Austin, TX, USA) to manage a constant current stimulator DS5 (Digitimer Ltd, Letchworth Garden City, UK) through a PC. This setup enables the activation of one of eight output channels of the multiplexer D188 (Digitimer Ltd, Letchworth Garden City, UK), facilitating the delivery of stimuli to the selected eight electrode pairs as desired.

Sixteen reusable self-adhesive TENS electrodes (axion GmbH, Leonberg, Germany, 25 mm × 40 mm) were paired and positioned along the centerline from the shoulder joint to the elbow of the right arm. Arranged circumferentially, each pair was spaced at intervals representing 1/8th of the arm circumference. One electrode of each pair was situated 5 mm above the centerline, while the other was positioned 5 mm below. These pairs were sequentially numbered (Fig 1), with pair 1 at the anterior, pair 3 at the lateral, pair 5 at the posterior, and pair 7 at the medial position of the arm. Adopting a vertical configuration with top and bottom electrodes yielded more tolerable stimulation compared to transverse placements, as observed in prior research [18], which also demonstrated varied perceptions based on electrode pair positions.

The biphasic rectangular current stimulation signal is characterized by its key parameters: amplitude $A$, pulse width $t_p$, pulse interval 1/pulse frequency $f_p$ and the number of pulses $n_p$.

## Thresholds, qualitative and spatial perceptions

Three amplitude thresholds [17] were determined for each participant in order to establish an operational range for an electrocutaneous warning system: (1) the perception threshold ($A_p$), denoting the minimum stimulus detectable by the participant; (2) the attention threshold ($A_a$),

indicating the stimulus intensity that draws attention; and (3) the intolerance threshold ($A_i$), representing the point at which the stimulus becomes intolerable.

To establish these thresholds, participants received a single biphasic stimulation pulse with a pulse width of 150 μs. The amplitude increased gradually from 0 mA to a maximum of 25 mA in adaptive steps of 0.1 mA, 0.2 mA, or 0.5 mA, chosen by a trained operator. Prior to the experiment, participants were briefed on the significance of the three thresholds, and the operator informed them which threshold would be assessed next to ensure participant focus.

In the study, muscle twitches were deemed undesirable. If observed, the current threshold determination halted, and the stimulation amplitude ($A$), known as the motor threshold, was noted. Consequently, these thresholds encompassed motor threshold values occurring before reaching the threshold under assessment. Additionally, the attention threshold indirectly gauged the intensity of attention-attracting stimulation.

At each threshold, the operator queried participants about their qualitative and spatial perceptions. Qualitative perception options included knocking, scratching, stinging, pain, muscle twitch, tickling, itching, pinching, or squeezing [17]. It's important to note that perceiving pain does not always align with reaching the intolerance threshold. Reaching intolerance means the stimulus is perceived as intolerable, while the qualitative perception is selected from the questionnaire and may include sensations like stinging, pinching, or pain. Painful perceptions could occur before the intolerance threshold was reached if the pain isn't intolerable. Spatial perception options covered locations between electrodes, at both electrodes, upper or lower electrode, extending beyond electrodes, and other body parts [17]. Participants were briefed on threshold definitions and questionnaire categories before the experiment, with any queries addressed beforehand.

## Experimental paradigm

The entire experiment lasted approximately 138±31 min (mean±standard deviation), encompassing preparation and all sub-experiments, all conducted on the same day. Duration primarily varied among participants. Mean durations±standard deviations for each sub-experiment are provided alongside their respective sub-headers. These durations incorporate preparation times for subsequent sub-experiments (around 2–4 minutes). The experiment took place at a room temperature of around 23˚C.

**Preparation (30±0 min).** The devices were turned on 30 minutes prior to the experiment for warming up. The stimulator DS5 was also stabilized using a load resistor (1 kΩ), connected to it while sending multiple stimulation pulses. To ensure a low impedance between the skin and electrodes, the participant's right upper arm was cleaned and moistened with a wet towel. Participants were instructed to sit comfortably, relax their arm, avert their gaze from the devices, and focus on stimulation perception. Throughout the experiment, the arm remained in a relaxed bent position on a table.

In the subsequent experimental sections, only electrode pair 3 (positioned laterally) was affixed to the participant's arm and utilized throughout the experiment, except for the spatial discrimination task around the arm. This decision was motivated by the observed minimal muscle twitching in this position (8).

**Reference threshold experiment (14±6 min).** This experiment was reported already in [17]. Because its values are used as a basis for the experiments reported in this paper, we repeat a short description of the reference threshold experiment. The threshold determination process was iterated at least three times until achieving three consecutive series with consistent thresholds, assessed by a trained operator. This repetitive process enabled participants to become accustomed to detecting thresholds and reporting qualitative and spatial perceptions

of the current stimulation. Mean values for perception, attention, and intolerance thresholds were calculated from the last three measurement series. For qualitative and spatial perception, the most frequently occurring perception from the three series was selected, with the last one chosen if variations were present.

A warning amplitude $A$ is defined as

$$A = A_a + 0.2(A_i - A_a), \tag{1}$$

that is calculated from the determined mean values of the attention threshold $A_a$ and the intolerance threshold $A_i$.

Next, another experiment (18±14 min) took place, which is reported in [17].

**Pulse intervals (11±6 min).**   At electrode pair 3, five consecutive pulses ($t_p$ = 150 μs, amplitude $A$) with varying pulse intervals (1/pulse frequency) were used for electrocuatenuous stimulation. At first, four example pulse intervals [18] were presented in order to give the participant a feeling about the temporal perceptions: 'Single pulses' (125 ms), 'Pulsating' (56 ms), 'Vibrating' (14 ms) or 'Continuous' (7 ms). Thereafter, the pulse interval range was sampled to capture the range of the temporal perceptions and the transition pulse intervals. Pulse intervals were decreased from 200 to 33 ms (26 steps), 25 to 7 ms (12 steps), 5 to 1 ms (9 steps), 0.7 ms and 0.5 ms, and the participants reported their temporal (single pulses, pulsating, vibrating, or continuous) and qualitative perceptions (knocking, scratching, stinging, pain, muscle twitch, tickling, itching, pinching, squeezing) for each stimulation. Additionally, the participants were asked if they felt a more inconvenient or weakened perception for each pulse interval. This experiment delivers values of the transition thresholds of the temporal perception changes.

**Amplitude variation (10±6 min).**   At electrode pair 3, we varied the amplitude of the stimulation and asked the participants to report their estimate of the presented amplitude. To discretize the amplitude range with sufficient detail to quantify the participants' accuracy while keeping the task feasible for a first-time user, the individual perception $A_p$ and intolerance $A_i$ thresholds have been used to generate 9 equally spaced values between $1.1 \cdot A_p$ and $0.9 \cdot A_i$, where each amplitude was assigned to the values 1 to 9. The stimulation signal consisted of 5 consecutive pulses with a pulse width of $t_p$ = 150 μs and a pulse interval of 333 ms. At first, all 9 amplitude steps were presented to the participant in a series of up and then down. After that, a few test runs were conducted while randomly presenting one of the nine amplitudes, asking for the perceived strength and revealing the actual strength thereafter. After this training phase, amplitudes were presented in the following order: 5-7-9-1-9-5-4-8-6-6-6-1-8-3-8-2-1-5-6-2-9-2-7-4-4-9-3-5-2-2-8-3-6-5-9-5-8-1-4-9-3-1-7-4-4-7-3-7-8-2-3-1-7-6 and the perceived amplitude reported by the participant was recorded. The amplitudes were given in the same non-randomized order to each participant to ensure each participant experienced the same amount and order of small and big stimulation amplitude changes. The amplitude sequence was chosen to include each amplitude six times and to provide similar quantities (3–4 instances) of the different amplitude step-sizes between successive stimuli, except for the most extreme step-sizes of +/-7 (2 instances each) and +/-8 (1 instance each), within a feasible sequence length. This supported the evaluation of the influence of the preceding stimulus on the perception of the current stimulus.

Next, one experiment (8±6 min) took place, which is not reported in this paper followed by two consecutive experiments (26±15 min and 20±9 min), which are reported in [17].

**Spatial discrimination around the arm (7±4 min).**   We stimulated at one of the eight electrode pairs and asked the participant which pair it was. The stimulation signal consists of five consecutive pulses with $t_p$ = 150 μs, amplitude $A$ and a pulse interval of 333 ms. At first, this signal was sent to each of the electrode pairs 1 to 8 and back to 1 while telling the

participant which electrode pair number was stimulated. After that, a few test runs were conducted while stimulating at one electrode pair position, asking for the perceived location and revealing the actual location thereafter. After this training phase, the electrode pairs were stimulated in the following order: 2-6-8-5-7-2-3-1-3-1-6-7-2-8-4-4-2-7-2-8-7-6-3-3-7-1-4-5-5-4-8-1-4-3-8-6-1-5-5-6 and the perceived location in terms of electrode pair number, as reported by the participant, was recorded. If muscle twitch was present at one of the electrode positions it was discarded from the sequence. The electrode pairs were given in the same non-randomized order to each participant in order to ensure each participant experienced the same amount and order of small and big stimulation location changes. The sequence was chosen to include each position five times and each relative difference of successive positions in approximately even quantity (4–6 instances each), within a feasible sequence length.

## Statistics and analysis

The experimental results have been analyzed using MATLAB 2023b® (The MathWorks, Inc., Natick, Massachusetts, USA).

The results of individual pulse intervals where the temporal perception changes its quality from 'Single pulses' to 'Pulsating', 'Pulsating' to 'Vibrating', and 'Vibrating' to 'Continuous' are visualized using line plots and are quantified by median $M$ and the interquartile range $IQR$ due to skewed distributions with outliers. Gender differences are quantified by a Wilcoxon test with $p$-values$< 0.05$ considered statistically significant.

The perception of varying amplitudes is quantified by the median $M$ and the interquartile range $IQR$, as the presented and reported amplitudes are on an ordinal scale from 1 to 9. In order to investigate the relation between the presented and reported amplitude, a linear mixed-effects model is defined as

$$r_{i,j} = \beta_0 + \beta_1 t_{i,j} + b_{0,i} + b_{1,i} t_{i,j} + \varepsilon_{i,j}, \tag{2}$$

the $j$-th reported amplitude $r$ of the $i$-th participant. The predictor variable $t$ denotes the true presented amplitude. The parameters $b_{0,i}$ denote the random deviation of the mean intercept $\beta_0$ and $b_{1,i}$ the random deviation of the mean slope $\beta_1$. Thus, each participant is represented by an individual intercept and slope, which are random and have a zero mean. The parameter $\varepsilon_{i,j}$ contains the unexplained variance.

In order to investigate whether the preceding amplitude influences the current reported amplitude, a mixed-effects model is defined as

$$\text{logit } \mathbf{P}\left(r_{i,j} < t_{i,j}\right) = \beta_0 + \beta_1 \cdot 1\left\{t_{i,j-1} \le t_{i,j}\right\} + \beta_2 t_{i,j} + b_{0,i} + b_{2,i} t_{i,j}. \tag{3}$$

The left side describes the logistic transformation $\text{logit}(p) = \log\frac{p}{1-p}$ of the probability $\mathbf{P}$ that the $j$-th reported amplitude $r_{i,j}$ of the $i$-th participant is smaller than the true presented amplitude $t_{i,j}$. The coefficients of the right side contain the fixed parameters: intercept $\beta_0$, the effect $\beta_1$ (on a logistic scale) if the predecessor amplitude $t_{i,j-1}$ is smaller or equal the true current amplitude $t_{i,j}$, the effect $\beta_2$ (on a logistic scale) of the true current amplitude $t_{i,j}$. Further, the random parameters of the deviation of the intercept $b_{0,i}$ and the effect $b_{2,i}$ of the true current amplitude of the $i$-th participant are included in the model.

The spatial discrimination of the stimulation is evaluated using a heat map. For each participant, the percentage of correctly perceived electrode pairs is determined for electrode pairs 1 to 8. The resulting percentages of the study group are quantified by mean and standard deviation. In order to compare the accuracy of spatial perception between the electrode pairs, Wilcoxon tests are used. The $p$-values are corrected by the Bonferroni-Holm procedure in order

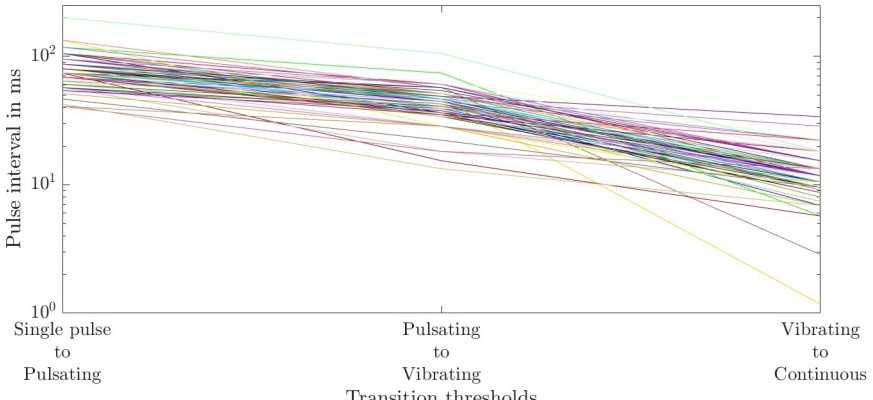

**Fig 2. Individual transition pulse intervals between the temporal perceptions.** 'Single pulse' to 'Pulsating' (blue), 'Pulsating' to 'Vibrating' (green), and 'Vibrating' to Continuous' (red). Each line represents one participant.

to control for the family-wise error rate. Corrected $p$-values $< 0.05$ are considered as statistically significant.

## Results

### Pulse intervals

Fig 2 shows the individual pulse intervals at which the temporal perception changes. The transition from 'Single pulse' to 'Pulsating' was perceived at a median [25%-75%] percentile of 74.1 [69.0–87.0] ms, from 'Pulsating' to 'Vibrating' at 44.4 [36.0–49.4] ms and from 'Vibrating' to 'Continuous' at 11.8 [9.5–15.4] ms. Around 49% of the participants reported a more inconvenient perception around 4.7 [3.6–7.8] ms compared to the other pulse intervals. For 67% of the participants, a weakened perception was present around the pulse intervals of 1.1 [1–1.3] ms. No significant difference in median values was found regarding gender.

### Amplitude variation

The median reported amplitudes [25%-75%-percentile] for presented stimuli with amplitudes 1 to 9 are: 1[1–1], 2[2–3], 3[2–4], 3[3–4], 4[3–5], 5[4–6], 6[4–7], 7 [5–8] and 7.5 [6–8]. We observe an underestimation of the perceived amplitude for presented amplitudes $\leq 4$ (Fig 3). Moreover, the violin plots in Fig 3 indicate a high variability of the perceived amplitudes.

Table 2 shows the result of the Wilcoxon test with the null hypothesis that the data in perceived amplitudes are observations from a distribution with medians 1 to 9, respectively. In order to treat the pseudoreplication due to the fact that each amplitude was presented six times to every participant, the individual mean perceived value was calculated out of 6. The null hypothesis was not rejected for amplitude 3 only.

Significant relationships ($< 0.05$) are marked in bold. $p$-values are corrected by the Bonferroni-Holm procedure.

A positive relationship between the presented and the reported amplitude is quantified by the linear mixed-effects model (cf. Table 3). The corresponding mean regression line is shown in Fig 3.

We observed a weak positive relationship between the predecessor amplitude onto the reported amplitude that is reflected in the fitting results of the mixed model (cf. Fig 4). The predictor that the predecessor amplitude is smaller or equal to the currently presented true

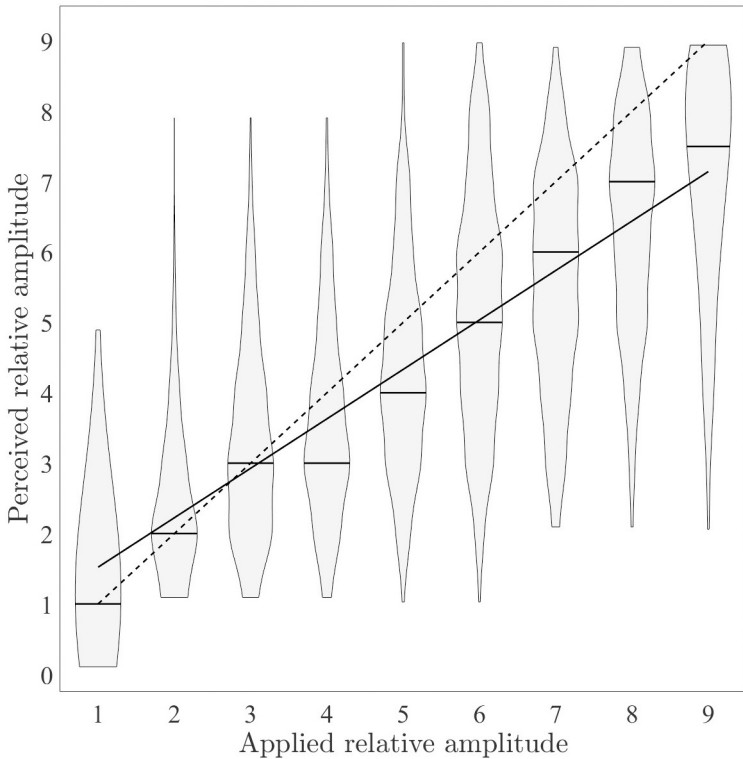

**Fig 3. Distributions of perceived relative amplitudes at applied relative amplitudes.** The dashed line represents the correct amplitude perception. The solid line represents the mean regression line of the linear mixed-effects model (Eq 2).

amplitude shows a statistically significant influence on the probability that the reported amplitude is smaller than the true amplitude. The corresponding parameter $\beta_1$ and its confidence interval do not overlap with zero. It can be observed that the intercept $\beta_0$ shows a larger variation (STD of $b_{0,i}$) than the slope $\beta_2$ (STD of $b_{2,i}$). The parameters $b_{0,i}$ and $b_{2,i}$ are negatively correlated.

Fig 5a shows that for female participants the median reported amplitude is correct from 1 to 6, whereas the reported amplitude is smaller than the presented one for amplitudes 4 to 9 for male participants. The Wilcoxon test indicates differing median values of the perceived amplitude for the presented amplitudes of 4 to 6, cf. Table 4.

## Spatial discrimination around the arm

Fig 6 shows the reported electrode pairs vs. the presented electrode pairs over all participants in %. The majority of the electrode pairs were reported correctly or at the two neighboring electrode pairs. Fig 6, upper bar, also shows the percentage of participants who showed muscle twitches at the corresponding electrode pairs, which led to the discarding of this pair for the spatial discrimination task. Muscle twitches occurred most often at the medial electrode

**Table 2. $p$-values of the Wilcoxon test with the null hypothesis that the data in perceived amplitudes are observations from a distribution with medians 1 to 9.**

| Amplitude | 1 | 2 | 3 | 4 | 5 | 6 | 7 | 8 | 9 |
|---|---|---|---|---|---|---|---|---|---|
| $p$-value | $\ll 0.05$ | 0.0024 | 0.0812 | $\ll 0.05$ | $\ll 0.05$ | $\ll 0.05$ | $\ll 0.05$ | $\ll 0.05$ | $\ll 0.05$ |

**Table 3. Parameters of linear mixed-effects model for the reported amplitude as the response variable and the presented amplitude as the predictor (Eq 2).**

| Parameter | $\beta_0$ | $\beta_1$ | STD of $b_{0,i}$ | STD of $b_{1,i}$ | STD of $\varepsilon_{i,j}$ |
|---|---|---|---|---|---|
| Estimate | 0.82 | 0.70 | 0.60 | 0.14 | 1.22 |
| Lower Limit | 0.66 | 0.67 | 0.48 | 0.11 | 1.19 |
| Upper Limit | 0.98 | 0.74 | 0.75 | 0.17 | 1.24 |

Estimates are given for the fixed parameters $\beta_0$ and $\beta_1$ and the standard deviations (STD) of the random parameters $b_{0,i}$, $b_{1,i}$ and the unexplained variance $\varepsilon_{i,j}$ with 95% confidence intervals (lower and upper limits of the covariance parameters).

positions 7 (36.7%) and 8 (59.2%). For electrode pairs 7 and 8, a higher muscle twitch frequency value is observable for males in Fig 7.

The percentages of correct perception of electrode pairs were calculated for each participant. The corresponding distributions were compared pair-wise between the electrode pairs. A rejection of the null hypotheses that the median values of two compared distributions are equal was observed for pairs: 1 vs. 4 ($p = 0.003$), 1 vs. 5 ($p = 0.037$) as well as 4 vs. 6 ($p = 0.028$), indicating a difference in perception accuracy for these pairs. The $p$-values were corrected using the Bonferroni-Holm procedure. Fig 7 shows the reported electrode pairs vs. the presented electrode pairs over all female (a) and male (b) participants in %. The heat maps indicate that the perception accuracy shows the largest variations between female and male participants for electrode pairs 1 and 4 (higher percentage values on the diagonal in males) and pair 6 (higher percentage values on the diagonal in females). From the percentages of correct electrode pair perceptions calculated for each participant, the corresponding distributions

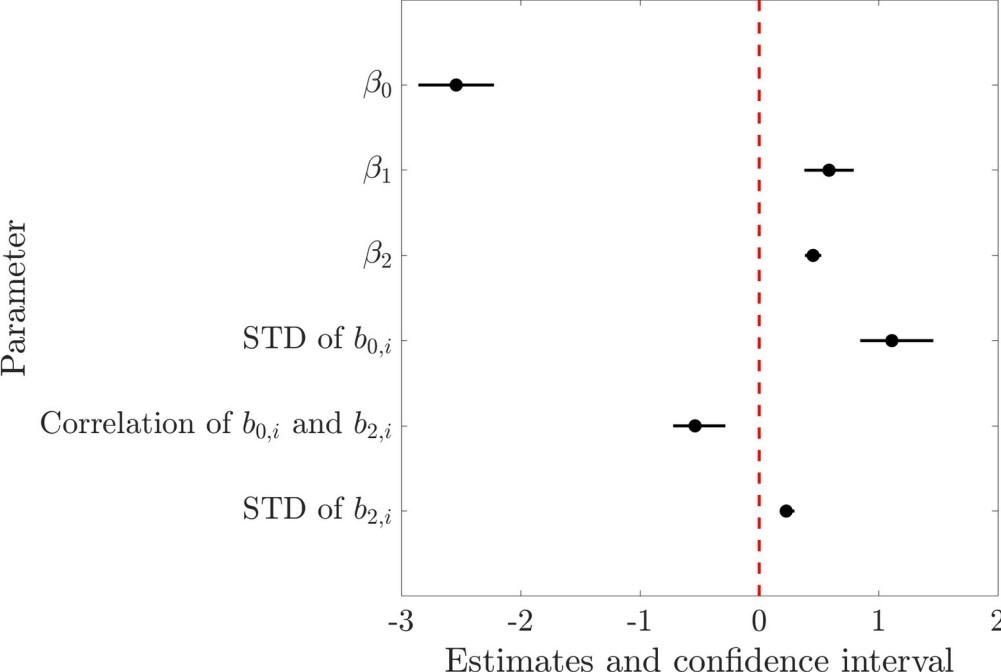

**Fig 4. Parameters of the mixed-effects model for the analysis of the predecessor amplitude on the reported amplitude (Eq 3).** Estimates are given for the fixed parameters $\beta_0$, $\beta_1$ and $\beta_2$ and the standard deviations (STD) of the random parameters $b_{0,i}$, $b_{2,i}$ as well as the correlation between and $b_{0,i}$ and $b_{2,i}$.

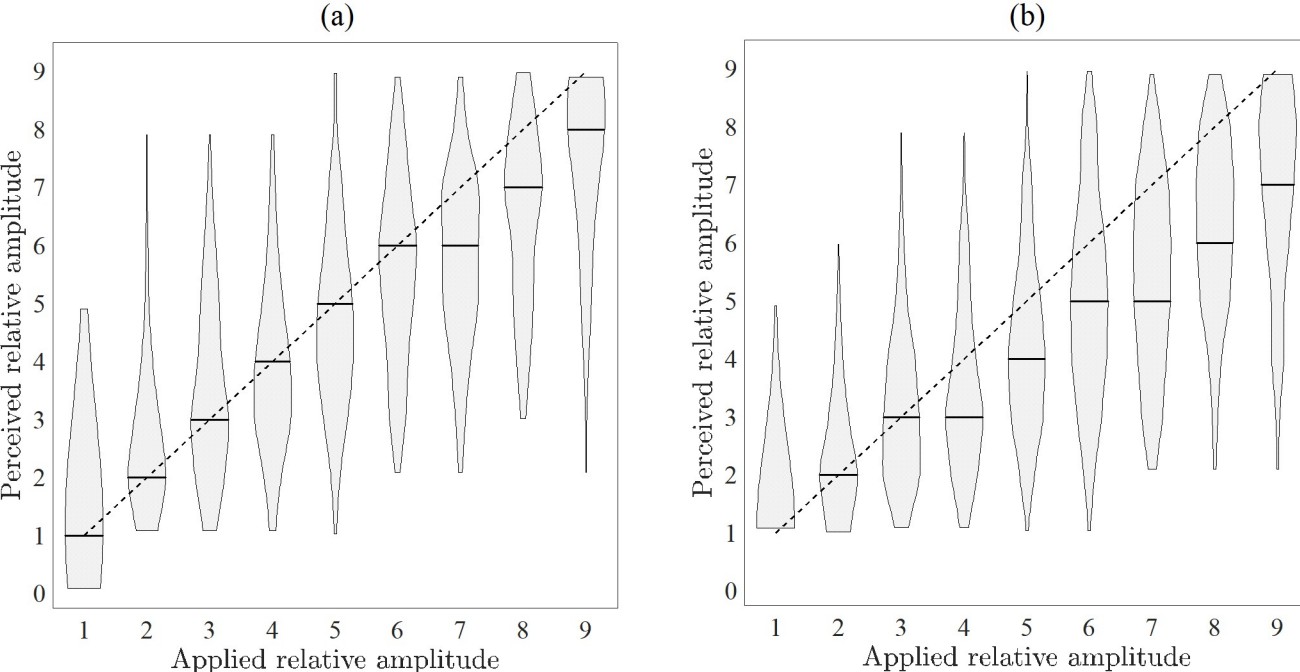

**Fig 5. Distributions of perceived relative amplitudes at applied relative amplitudes in dependence of the gender.** (a) Female, (b) Male. The dashed line represents the correct amplitude perception.

were compared pair-wise between the electrode pairs separately for both genders. For the female subgroup, a rejection of the null hypotheses that the median values of two compared distributions are equal was observed for pairs: 4 vs. 5 ($p = 0.041$) and 4 vs. 6 ($p = 0.007$). For the male subgroup, a rejection of the null hypotheses that the median values of two compared distributions are equal was observed for pairs: 1 vs. 3 ($p = 0.007$), 1 vs. 4 ($p = 0.0024$), and 1 vs. 5 ($p = 0.019$). The $p$-values have been corrected using the Bonferroni-Holm procedure.

## Discussion

We found that the thresholds for the transition of the temporal perception from 'Single pulses' to 'Pulsating' to 'Vibrating' and 'Continuous' show a high interindividual variance (Fig 2). Such high interindividual variability might be partly explained by the involvement of subjective judgment related to perceptual and attitudinal traits in individuals [23]. Consequently, an individual determination of the intervals might be advisable for the design of a potential warning signal. However, the median pulse intervals found in this study can serve as an initial guess for the individual determination. For example, if a warning signal should be of type 'Vibrating', a pulse interval of 28 ms would be the initial guess for searching for the optimal individual pulse interval to elicit a vibrating sensation. We found no statistically significant differences in the transition thresholds of the temporal perception regarding gender. Geng et al. [24] found that women showed better abilities to distinguish stimulation intervals. Gender differences have been noted also in other studies [25–27]. Consequently, future studies and the design of potential warning signals should take into account the possibly different perceptions of men and women in warning signals.

The stimulation with amplitude step 1 showed the smallest *IQR* of the reported amplitudes with [1–1]. Due to the fact that this presented amplitude is only 1.1 times larger than the

**Table 4. Comparison between genders for perceived amplitudes by Wilcoxon test for presented amplitudes 1 to 9.**

| Amplitude | 1 | 2 | 3 | 4 | 5 | 6 | 7 | 8 | 9 |
|---|---|---|---|---|---|---|---|---|---|
| *p*-value | 0.933 | 0.168 | 0.051 | **0.027** | **0.039** | **0.033** | 0.188 | 0.429 | 0.070 |

*p*-values<0.05 indicate a significant difference.

perception threshold $A_{\mathrm{P}}$, 8% of the participants reported that they could barely feel the stimulation, thus it was easier for them to identify amplitude step 1. For presented amplitude steps ≥4, the median perceived amplitude was around one amplitude step smaller than the presented one. A possible explanation for the underestimation of the presented amplitude might be the difference in attention as stimuli 1–2 are below and stimuli 4–9 are above the attention threshold. On the ordinal scale, the attention threshold lies at step 3 (median value over all participants). Attention has been suggested to modulate the perception of noxious stimuli [28], tactile stimuli [29], and auditory stimuli [30]. In comparison between the genders, this effect was only visible for male participants (Fig 6b). The median perceived amplitude was smaller than the presented one for female participants for presented amplitude steps ≥7. This difference is reflected for presented amplitudes 4, 5, and 6 (cf. Table 4) by the rejection of the null hypothesis that perceived amplitudes from female participants and perceived amplitudes from male participants are samples from continuous distributions with equal medians. A possible partial explanation for the gender difference might be the larger female sensitivity for electrical stimuli applied at the sensory level [25] leading more often to an accurate perception of the presented amplitude. The analysis of the effects of the predecessor amplitude on the perceived amplitude of the current stimulus revealed that if the predecessor amplitude is smaller than the current amplitude, the probability of underestimating the true amplitude is increased. Further,

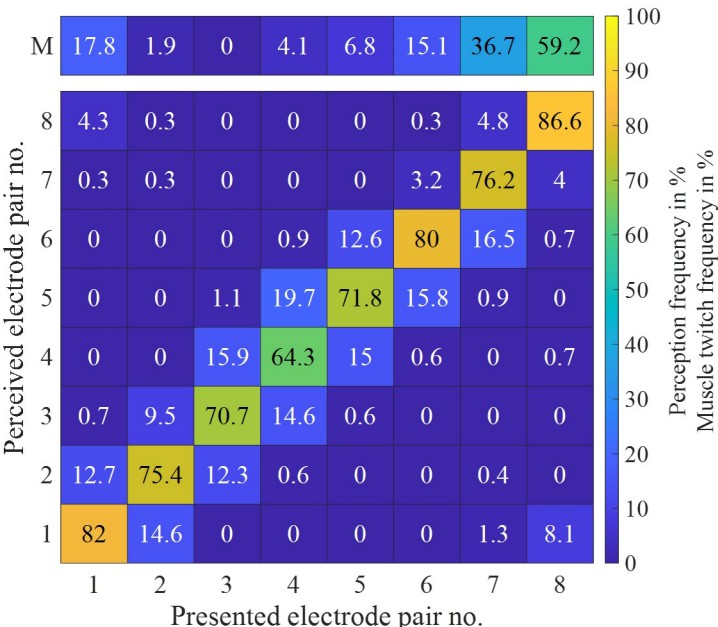

**Fig 6. Reported electrode pair vs. presented electrode pair in %.** The upper line indicates how many participants experienced muscle twitches at the corresponding electrode pair, which led to rejection of this pair in the stimulation series.

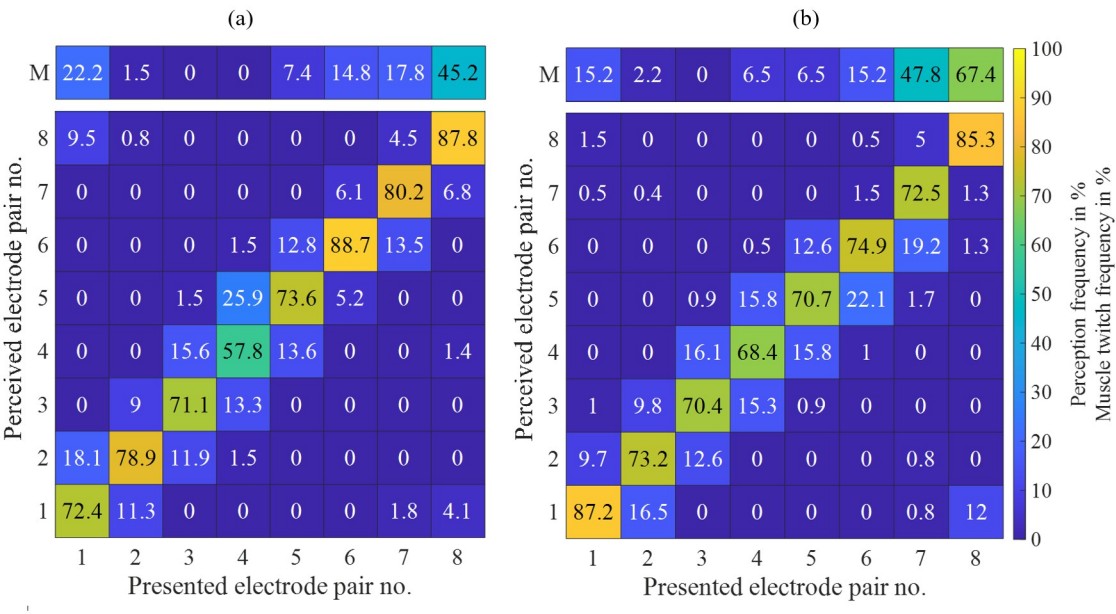

**Fig 7. Reported electrode pair vs. presented electrode pair in % in dependence of the gender.** (a) Female, (b) Male. The upper line indicates how many participants experienced muscle twitches at the corresponding electrode pair, which led to rejection of this pair in the stimulation series.

the variances of the amplitude estimation (Fig 3) lead us to the conclusion that the fine-grained amplitude variations of individual pulses are of limited use for direct information encoding in warning signals. Our finding is supported by studies in the field of sensory feedback for prosthesis, where the widely used parameters to encode sensory feedback are e.g. pulse frequency [19,20], number of pulses [21], and pulse width [19] whereas a fixed amplitude is often used. However, the results indicate a linear relationship between presented and perceived amplitude, which may be incorporated in structured stimulation patterns.

In the spatial discrimination task, the majority of the electrode pair locations were reported correctly (cf. Fig 7). However, it should be taken into account that due to muscle twitches, fewer channels were present for some participants (cf. Fig 7). It cannot be ruled out that the task became easier with the omitted channels because, for example, two fewer channels had to be localized. The highest correct perception frequency was found at electrode pair 8, but due to the amount of muscle twitching (cf. Fig 7) only 41.8% of participants reported the localization for this pair. In part 1 of this experiment, we report that muscle twitches in the used setup are unavoidable suggesting it is unsuitable for warning workers during fine motor tasks [17]. The occurrence and intensity of muscle twitches during electrical stimulation are influenced by individual anatomical factors, such as muscle length, with longer muscles producing greater torque during stimulation [6]. These individual variations should be taken into account in the continued development of the electrical warning system. However, this limitation is probably less important for the typical application scenarios with workers on railway tracks or similar, where minor muscle twitches might be tolerable during a warning. The varying arm circumferences of the participants led to a varying spacing between neighboring electrode pairs. Moreover, we used a fixed electrode size for the varying arm circumferences, which might stimulate a larger fraction of the receptive skin surface for smaller arm circumferences compared to larger arm circumferences. Pearson's correlation coefficient between the arm

circumference and the detection rate was $r = 0.29$ ($p = 0.01$). Thus, it was easier for participants with larger arm circumferences to distinguish the stimulation location. The results regarding gender indicate a lower spatial perception accuracy at electrode pair 4 (lateral-posterior) for female participants and an increased spatial perception accuracy for male participants at electrode pair 1 (anterior). This might be partially explained by the fact that the amount of subcutaneous fat tissue is higher in women compared to men.

We assume that the exact localization of the stimulus is not necessary for a warning signal, but the discriminability between different electrode positions is of interest for the design of a warning signal. In this study, 5 consecutive pulses with a pulse interval of 333 ms were used to discriminate between different stimulation locations. Boldt et al., 2014 [22] reported that the two-point discriminability depends on temporal stimulation parameters like the number of pulses, where more pulses lead to better discriminability. This can be taken into consideration for the design of a warning signal. Geng et al., 2014 [21] showed that spatial identification worked better if only one electrode pair was stimulated instead of two simultaneously. They concluded that paired parameters work worse for spatial discrimination.

There are some limitations to our study. The investigated electrode configurations were limited to vertical pairs and a fixed electrode size, which was based on practical considerations. Other configurations like transversal or diagonal electrode pairs or the influence of the neighboring margin lengths of the electrodes should be investigated in future studies. The experimental setup used in this study was designed primarily for baseline research in the laboratory. However, its stationary nature restricts its use in field studies and real-world warning scenarios. Future work will focus on miniaturizing the setup to make it portable, with the long-term goal of creating a wearable version for users. Further, the experiments have been conducted at the lateral electrode pair no. 3 with a pulse width of 150 µs (except for the spatial discrimination task). Although future studies need to investigate the effects of pulse width for all electrode positions, we believe that a pulse width of 150 µs can be chosen as a basic working parameter for a future warning system [17]. TENS electrodes were selected for this study due to their low and consistent impedance characteristics. However, they are not ideal for long-term use, as the gel can dry out over time, reducing their adhesive properties. In future work, textile electrodes will be further explored as a more suitable solution for wearable applications. In this study, a straightforward method of consecutively increasing the stimulation amplitude was used to determine perception, attention, and intolerance thresholds. This approach was chosen over more precise alternatives, such as the best PEST method [31], to maintain a manageable experimental duration and minimize participant fatigue or loss of concentration. The experimental paradigm was presented to all participants in the same order to ensure consistency across individuals. However, the absence of randomization may have introduced sequence effects. Gender differences reported here should be considered with caution since we had unequal sample sizes (29 females and 52 males). Young participants dominated the age distribution of the study group. Only 6 participants were older than 40 years. Consequently, future studies need to address the influence of age on the thresholds. Future studies will try to provide a more uniform distribution of participants between 18 and 65 years of age as other studies of electrocutaneous stimulation showed age-related effects on the perception threshold [26,32]. The majority of participants in this study were university students. Future research will focus on conducting experiments with employees, who represent potential users of an electrical warning system. Another limitation of this study is the lack of blinding, as the same individuals were involved in both conducting the experiments and evaluating the results, at least in part. In future work, these tasks will be separated to introduce a degree of blinding and reduce potential bias.

## Conclusions

Our study contributes stimulus interval values for the transition of the temporal perception between Pulsating, Vibrating, and Continuous. While these perceptions can be produced as needed and in this order across the pulse interval range, we found a high interindividual variability of the absolute pulse interval values. Therefore, the absolute values should be calibrated to the individual using the distributions reported in this study as a default starting point.

We established a linear relationship between presented and perceived amplitudes of the electric warning signals. This relationship might be used in structured warning signals and can serve as one means of coding information into this warning signal.

We established the discriminability of electric warning stimulation locations around the arm. This opens the possibility of encoding information about the type of warning into the signal in a workplace safety device. Given the high discriminability accuracy when taking the neighboring electrode pairs into account, four principal locations might be used for this information encoding.

In summary, this study underlined the realizability of an electric warning system for workplace safety and explored design dimensions of electrocutaneous stimulation signals at the upper arm with a biphasic current stimulation pulse during rest. For the design of an electrical warning signal, varying pulse intervals and stimulation locations (i.e. electrode pairs around the upper right arm) are directly applicable, while the stimulation amplitude may be incorporated in a structured stimulation pattern. Our research lays the groundwork for developing suitable signals for wearable electric warning devices.

## Acknowledgments

We thank Stefan Heyder of the TU Ilmenau statistical consulting service for discussions about the statistical evaluation.

## Author Contributions

**Conceptualization:** Stephan Lau, Jens Haueisen.

**Formal analysis:** Stephan Lau, Jens Haueisen.

**Funding acquisition:** Eva-Maria Dölker, Stephan Lau, Jens Haueisen.

**Investigation:** Eva-Maria Dölker, Stephan Lau, Jens Haueisen.

**Methodology:** Stephan Lau, Maria Anne Bernhard, Jens Haueisen.

**Project administration:** Eva-Maria Dölker, Stephan Lau, Jens Haueisen.

**Software:** Stephan Lau.

**Supervision:** Jens Haueisen.

**Validation:** Eva-Maria Dölker, Jens Haueisen.

**Visualization:** Eva-Maria Dölker, Maria Anne Bernhard.

**Writing – original draft:** Eva-Maria Dölker.

**Writing – review & editing:** Eva-Maria Dölker, Stephan Lau, Maria Anne Bernhard, Jens Haueisen.

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
