## [Decision Letter · Decision Letter 0]

18 Jul 2024

PONE-D-24-17781Design dimensions of electrocutaneous warning stimulation patterns: pulse intervals, amplitude, and locationPLOS ONE

Dear Dr. Dölker,

Thank you for submitting your manuscript to PLOS ONE. After careful consideration, we feel that it has merit but does not fully meet PLOS ONE’s publication criteria as it currently stands. Therefore, we invite you to submit a revised version of the manuscript that addresses the points raised during the review process.

We look forward to receiving your revised manuscript.

Kind regards,

Hadeel K. Aljobouri

Academic Editor

PLOS ONE

 [copy in funding statement].  

4. We noted in your submission details that a portion of your manuscript may have been presented or published elsewhere. [Yes, In this paper, we report the second part of a set of experiments from a study on 81 volunteers where the first part is already published in 10.1038/s41598-022-10708-9.] Please clarify whether this publication was peer-reviewed and formally published. If this work was previously peer-reviewed and published, in the cover letter please provide the reason that this work does not constitute dual publication and should be included in the current manuscript.

5. Please provide a complete Data Availability Statement in the submission form, ensuring you include all necessary access information or a reason for why you are unable to make your data freely accessible. If your research concerns only data provided within your submission, please write "All data are in the manuscript and/or supporting information files" as your Data Availability Statement.

Reviewers' comments:

Reviewer's Responses to Questions

**Comments to the Author**

1. Is the manuscript technically sound, and do the data support the conclusions?

Reviewer #1: Yes

Reviewer #2: Partly

Reviewer #3: Yes

2. Has the statistical analysis been performed appropriately and rigorously? 

Reviewer #1: Yes

Reviewer #2: Yes

Reviewer #3: N/A

3. Have the authors made all data underlying the findings in their manuscript fully available?

Reviewer #1: Yes

Reviewer #2: Yes

Reviewer #3: Yes

4. Is the manuscript presented in an intelligible fashion and written in standard English?

Reviewer #1: Yes

Reviewer #2: Yes

Reviewer #3: Yes

5. Review Comments to the Author

Reviewer #1: Overall, this is a clear, manuscripte, and well-written manuscript. ...............................................................................................................................................................................................................................................................................................................................................................................................................................................

Reviewer #2: 1. The introduction must highlight the main electrical stimulation basics and occupational safety.

2. The contribution of the proposed work should be presented.

3. The applicable guidelines and regulations the authors refer to in the Methodology Section need some focus, and the references related to these regulations must be inserted.

4. In the experimental work section, a block diagram and paragraph about the work should be added even if the author refers to previous relevant research

5. As the presented work is essential in occupational safety, what are the main challenges for the proposed work?

6. The conclusion needs an improvement.

Reviewer #3: The manuscript titled "Design Dimensions of Electrocutaneous Warning Stimulation Patterns" addresses an important topic and presents valuable insights into the design dimensions of electrocutaneous warning stimulation patterns. With some revisions to enhance clarity, provide additional context, and strengthen the discussion, this study has the potential to make a significant contribution to the field.

1. The title effectively conveys the focus of the study on electrocutaneous warning stimulation patterns. However, it could be improved by specifying the specific context or application of these patterns to enhance clarity for readers.

2. The abstract provides a concise summary of the study objectives, methods, and key findings. It effectively highlights the importance of the research but could benefit from more specific details on the experimental design and implications of the results.

3. The introduction provides a clear rationale for the study by discussing the importance of optimizing electrocutaneous warning stimulation parameters for occupational safety. However, it would be beneficial to include more background information on existing research in this area to better contextualize the current study.

4. The methodology section is detailed and well-structured, providing a clear overview of the experimental design, participant characteristics, and data analysis procedures. However, additional information on the selection criteria for participants and the rationale behind the chosen stimulation parameters would enhance the methodological clarity.

5. The results section presents the findings of the study in a logical sequence with appropriate statistical analysis. It effectively communicates the outcomes of varying pulse intervals, amplitudes, and electrode pair positions on temporal perception. Including visual aids such as tables or figures to supplement the text would further enhance the presentation of results.

6. The discussion interprets the results in the context of the study objectives and compares them to existing literature. It effectively addresses the implications of the findings for occupational safety applications. However, expanding on the limitations of the study and suggesting potential avenues for future research would strengthen the discussion section.

7. The conclusion succinctly summarizes the key findings and their significance for optimizing electrocutaneous warning stimulation patterns. It effectively reinforces the study's contributions to the field of occupational safety.

8. It is essential to ensure that all ethical considerations, such as approval from relevant ethics committees and informed consent from participants, are clearly documented in the manuscript to uphold research integrity and compliance with ethical standards.

6. PLOS authors have the option to publish the peer review history of their article (what does this mean?). If published, this will include your full peer review and any attached files.

Reviewer #1: **Yes: **Ahmed S.Alghabban

Reviewer #2: **Yes: **Amal Ibrahim Mahmood

Reviewer #3: No

---

## [Author Response · Author response to Decision Letter 0]

27 Sep 2024

Rebuttal letter

Dear Dr. Hadeel K. Aljobouri and Reviewers,

On behalf of my co-authors and myself, I extend our sincere thanks for your time and effort in reviewing our manuscript titled “Design dimensions of electrocutaneous warning stimulation

patterns: pulse intervals, amplitude, and location”. Your detailed comments and constructive feedback have been very helpful in improving our work. We appreciate your thorough

evaluations and professional guidance. Please find our responses to your comments in the attached file rebuttal letter.pdf

Best regards,

Dr. Eva-Maria Dölker

Institute for Biomedical Engineering and Informatics

Technische Universität Ilmenau

98693 Ilmenau

eva-maria.doelker@tu-ilmenau.de

---

## [Editor Report · Decision Letter 1]

14 Oct 2024

PONE-D-24-17781R1Design dimensions of electrocutaneous warning stimulation patterns in workplace safety devicesPLOS ONE

Dear Dr. Dölker,

Thank you for submitting your manuscript to PLOS ONE. After careful consideration, we feel that it has merit but does not fully meet PLOS ONE’s publication criteria as it currently stands. Therefore, we invite you to submit a revised version of the manuscript that addresses the points raised during the review process. Please submit your revised manuscript by Nov 28 2024 11:59PM. If you will need more time than this to complete your revisions, please reply to this message or contact the journal office at plosone@plos.org. Please include the following items when submitting your revised manuscript:A rebuttal letter that responds to each point raised by the academic editor and reviewer(s). You should upload this letter as a separate file labeled 'Response to Reviewers'.A marked-up copy of your manuscript that highlights changes made to the original version. You should upload this as a separate file labeled 'Revised Manuscript with Track Changes'.An unmarked version of your revised paper without tracked changes. You should upload this as a separate file labeled 'Manuscript'.If applicable, we recommend that you deposit your laboratory protocols in protocols.io to enhance the reproducibility of your results. Protocols.io assigns your protocol its own identifier (DOI) so that it can be cited independently in the future. For instructions see: https://journals.plos.org/plosone/s/submission-guidelines#loc-laboratory-protocols. Additionally, PLOS ONE offers an option for publishing peer-reviewed Lab Protocol articles, which describe protocols hosted on protocols.io. Read more information on sharing protocols at https://plos.org/protocols?utm_medium=editorial-email&utm_source=authorletters&utm_campaign=protocols.

We look forward to receiving your revised manuscript.

Kind regards,

Hadeel K. Aljobouri

Academic Editor

PLOS ONE

Journal Requirements:

Additional Editor Comments:

• Keywords must be included in the manuscript.

• The number of references is minimal, they must be 30 minimum. We prefer to add more regarding the simulation, I recommend the authors read this paper as an example:

“A Virtual EMG Signal Control and Analysis for Optimal Hardware Design,” International Journal of Online and Biomedical Engineering (iJOE), vol. 18, no. 02, pp. 154–166, Feb. 2022, doi: 10.3991/IJOE.V18I02.27047.

However, there is no need to cite the paper.

---

## [Author Response · Author response to Decision Letter 1]

28 Oct 2024

Dear Dr. Hadeel K. Aljobouri and Reviewers,

On behalf of my co-authors and myself, I extend our sincere thanks for your time and effort in reviewing our manuscript titled “Design dimensions of electrocutaneous warning stimulation patterns in workplace safety devices”. Your detailed comments and constructive feedback have been very helpful in improving our work. We appreciate your thorough evaluations and professional guidance.

Please find our responses in the attached rebuttal letter.

Best regards,

Eva-Maria Dölker

---

## [Editor Report · Decision Letter 2]

13 Nov 2024

Design dimensions of electrocutaneous warning stimulation patterns in workplace safety devices

PONE-D-24-17781R2

Dear Dr. Dölker,

We’re pleased to inform you that your manuscript has been judged scientifically suitable for publication and will be formally accepted for publication once it meets all outstanding technical requirements.

Kind regards,

Hadeel K. Aljobouri

Academic Editor

PLOS ONE
---

## [Editor Report · Acceptance letter]

5 Dec 2024

PONE-D-24-17781R2 

PLOS ONE

Dear Dr. Dölker, 

I'm pleased to inform you that your manuscript has been deemed suitable for publication in PLOS ONE. Congratulations! Your manuscript is now being handed over to our production team.

Kind regards, 

on behalf of

Asst.Prof.Dr. Hadeel K. Aljobouri 

Academic Editor

PLOS ONE